# Contrasting Flexible and Rigid Bioinspired Flapping Hydrofoils for Suspended Particles Discharge in Raceway Aquaculture

**DOI:** 10.3390/biomimetics10110779

**Published:** 2025-11-16

**Authors:** Fangwei Xu, Ertian Hua, Mingwang Xiang

**Affiliations:** The College of Mechanical Engineering, Pingfeng Campus, Zhejiang University of Technology, Hangzhou 310007, China; 2111915045@zjut.edu.cn (F.X.); 211122020035@zjut.edu.cn (M.X.)

**Keywords:** bioinspired flapping hydrofoils, two-way fluid–structure coupling, raceway aquaculture, suspended particles

## Abstract

To investigate the impact of flexible versus rigid bioinspired flapping hydrofoils on the discharge characteristics of suspended particles in raceway aquaculture, this study established a two-way fluid–structure coupling model of a flapping hydrofoil device based on ANSYS Fluent and Transient Structural modules. The research compares the discharge characteristics of hydrofoils with different elastic moduli. The results show that, within a certain range of elastic moduli adjustment, flexible bioinspired hydrofoils exhibit greater surface deformation compared to rigid ones, effectively delaying tail vortex shedding and extending its duration, thus prolonging the range of high flow velocities. During the middle stage of discharge, the escape rate of suspended particles under the influence of flexible bioinspired hydrofoils with 0.05 GPa elastic modulus was 3–4% higher than that of rigid hydrofoils. However, in terms of achieving maximum discharge efficiency and effectiveness, both reached approximately 97.8% with little difference between them. This study highlights the bioinspired principles in hydrofoil design and provides a reference for optimizing flexible hydrofoil discharge characteristics in future research.

## 1. Introduction

In-Pond Raceway System (IPRS) is a new type of high-density aquaculture model [1,2]. This model innovates the traditional “open-pond” approach into a new “enclosed” circulating water system. The integrated aquaculture system allocates 3–5% of the total water area for production units featuring water circulation, oxygenation, and waste extraction mechanisms, with the dominant 95–97% area functioning as a bioremediation zone for effluent treatment [3]. The overall layout is shown in Figure 1. This aquaculture model, centered on high-density farming in small areas and water quality purification in large areas, significantly reduces water resource waste through the recycling of aquaculture water, offering advantages of high productivity, water conservation, and environmental protection [4].

IPRS can be implemented in various forms. When applied to fish aquaculture, the main components of suspended particles in the water are residual feed and fish feces [5,6,7], which are the primary factors affecting water quality. These suspended particles, with densities like water, can damage fish gill structures and affect fish health and quality [8,9]. Additionally, studies have shown that water flow in aquaculture is crucial for improving fish survival rates and growth [10], affecting not only the distribution of dissolved oxygen, organic and inorganic substances, and other nutrients in the water but also the self-cleaning properties of aquaculture systems [11]. Therefore, enhancing the targeted migration efficiency of suspended particles by improving aquaculture water flow is one of the core challenges in optimizing IPRS.

Addressing this issue, experimental investigations by Li et al. [12] demonstrated the constraints of conventional aeration piston flow devices (APFDs) in raceway systems. Their study identified turbulent flow patterns and centripetal effects generating distinct flow irregularities—including localized plug flow and reverse currents—along the raceway’s horizontal plane. These hydrodynamic interactions contribute to the retention and buildup of contaminants within the system. Wang Xiang et al. [13] simulated flow velocities in aquaculture systems using a dense discrete phase model, conducting numerical simulations of suspended solid particles with different diameters. The research showed that solid-phase particles exhibit a “U-shaped” sedimentation distribution in the pollution collection area of the raceway. Wu Yanfei et al. [14] used physical experiments to study the impact of submersible propeller deployment on pollutant accumulation in octagonal aquaculture ponds. The research indicated that diagonal deployment offers superior pollution collection performance in octagonal ponds, providing a valuable reference for optimizing the pollution discharge performance of raceway aquaculture systems. Currently, in practical applications, many domestic companies have designed and produced various specifications of water-pushing aeration equipment, but most aeration devices are surface aerators, such as impeller, microbubble or nanobubble, and surge-type aeration equipment. The dissolved oxygen content in water can be effectively elevated by these mechanisms, resulting in enhanced water quality. But their enhancement of deep-water flow is limited, restricting the overall efficiency of pollutant discharge from the water body [15].

Research has found that fish-tail-mimicking flapping hydrofoil devices can effectively enhance water flow [16], and the induced reversed Kármán vortex street can improve issues such as insufficient water hydrodynamics and particle accumulation in waterways. Current research on these flapping devices primarily focuses on the optimization of motion parameters and structural parameters of rigid flapping hydrofoils, with relatively few studies on their application in enhancing the self-cleaning of aquaculture systems. In terms of optimizing the motion and structural parameters of flapping devices, Ma Penglei et al. [17] proposed an improved model based on the traditional motion model, analyzing the hydrodynamic performance and energy conversion efficiency of the improved model under different motion parameters. Sun et al. [18] proposed an oscillating hydrofoil model with a trailing edge flap, using CFD software’s dynamics and dynamic mesh technology to numerically simulate a simple oscillating hydrofoil structure model. Hu Feng et al. [19] analyzed the propulsive forces generated by hydrofoils during movement, exploring the influence of wave parameters (wave height, period) and structural parameters (aspect ratio) on the propulsive performance of hydrofoils.

In research applying flapping hydrofoil devices to aquaculture system pollutant discharge, Hua Er researchers analyzed the relationship between flapping device deployment depth, frequency of oscillation, and the resulting displacement of waterborne suspended particles. Simulation results showed that a suitable combination of installation depth and oscillation frequency can effectively promote more efficient particle clearance [20]. Additionally, this team conducted a systematic comparative analysis of the channel characteristics and pollutant discharge performance of hydrofoils with different chord lengths and frequencies, providing recommendations for the optimal combination of hydrofoil chord length and oscillation frequency for pollutant discharge performance [21].

Furthermore, hydrofoil deformation is common in practical application scenarios, and research has shown that flexible hydrofoils can provide advantages of high lift and thrust [22]. David et al. [23] measured the thrust coefficient of a flexible panel installed on the trailing edge of a rigid hydrofoil, finding that at specific bending stiffness and pitch frequency, the thrust of the flexible hydrofoil was 2–4 times that of the rigid one. Wang Qiang et al. [24], using a two-way fluid–structure coupling method, found that flexible hydrofoils significantly improved propulsion performance during flapping. Wang Qi et al. [25] discovered that compared to rigid airfoils, flexible airfoils delayed the time of tail vortex shedding, effectively reducing the oscillation frequency of lift disturbances. These studies provide important references for further optimizing hydrofoil performance parameters and promoting the practical application of flapping hydrofoil devices.

With the evolution of numerical computing, Computational Fluid Dynamics (CFD) simulation experiments offer greater flexibility and cost-effectiveness compared to traditional physical experiments. Using this technology to explore the removal of solid particulate matter from aquaculture ponds can effectively support aquaculture system research [26,27,28].

Based on the above discussion, rectangular raceway culture systems serve as the exemplary configuration in this investigation and employs two-dimensional numerical simulation methods with ANSYS-Fluent 2022R2 to establish a two-way fluid–structure coupling model. The aim is to explore the influence of material properties on the pollution discharge efficiency of flexible hydrofoils for suspended particulate pollutants in aquaculture tanks, providing reference and guidance for further improving the removal efficiency of suspended particles in the aquaculture environment. To clarify the nature of this study, this work is a numerical investigation. We solve a two-way fluid–structure interaction (FSI) problem using ANSYS Fluent coupled with Transient Structural to compare flexible and rigid bioinspired flapping hydrofoils in an in-pond raceway configuration. No new physical experiments were conducted; literature data are used only for validation.

## 2. Physical Model

### 2.1. Raceway Aquaculture Model

IPRS typically includes aquaculture and water maintenance areas. To obtain uniform water flow characteristics, the aquaculture area raceway generally adopts a rectangular structure [29]. As shown in Figure 2, the raceway dimensions selected in this study are 21 m in length, 5 m in width, and 2 m in water depth. In the length direction of the raceway (x-direction), to fully utilize the operational efficiency of the flapping hydrofoil, the pivot position of the hydrofoil is fixed at 1 m from the raceway inlet. The z-coordinate pumping depth represents the vertical span from the water’s surface to the hydrofoil’s motion center. Research findings [30] reveal that near-wall positioning of the oscillating hydrofoil creates substantial wall interference, leading to spatial redistribution of the reverse Kármán vortex wake. This situation can lead to uneven distribution of pollutants, affecting the overall discharge efficiency. Additionally, research has indicated that excessive pumping depth will cause instability in the generated reversed Kármán vortex street, reducing particle discharge efficiency, with a pumping depth of 1.1 m showing better particle escape rates than other working conditions [20]. However, during actual simulation, it was found that, unlike rigid hydrofoils, the deformation of flexible hydrofoils might lead to contact between the hydrofoil and the wall. Considering both wall effects and vortex street stability, a pumping depth of 1.0 m was adopted in subsequent research.

### 2.2. Hydrofoil Motion Model

The oscillating motion of the hydrofoil, inspired by the flapping wings of birds and fish fins, mimics the natural fluid dynamics processes observed in nature. This bioinspired motion enables the hydrofoil to adapt to varying flow conditions, enhancing the system’s efficiency in pollutant removal. The motion pattern of the hydrofoil aims to emulate the adaptive and flexible movements of biological wings or fins, which interact with the fluid environment to optimize propulsion and flow dynamics.

The experimental setup employed NASA’s NACA0012 symmetric hydrofoil from the NACA series as the active element in the flapping foil apparatus. The hydrofoil chord length c = 1 m, with the pivot position set at 0.2 c (0.2 m) from the leading edge. Research has shown that this position can achieve optimal water propulsion efficiency for the flapping hydrofoil [31]. Figure 3 shows the dimensions of the selected hydrofoil.

The hydrofoil’s motion includes heaving and pitching, adopting a conventional sinusoidal motion model [32]. In the two-way fluid–structure coupling, the flapping motion was defined in the Transient Structural module using analytical functions. The basic equations of motion are as follows:(1)$$\left\{\begin{array}{l} y(t)=A_{\max}\sin(2\pi f)\\ \theta (t)=\theta_{\max}\sin(2\pi f+\varphi ) \end{array}\right.$$
where $A_{\max}=0.5\;m$ is the maximum heaving amplitude of the flapping hydrofoil; This value was chosen to ensure a sufficient gap between the hydrofoil and the bottom wall during motion while keeping the flow in an efficient jetting state. Ding H et al. [33] found that when $\theta_{\max}$ exceeds 30°, the water propulsion efficiency of the flapping hydrofoil begins to decrease. To maximize the propulsion efficiency, $\theta_{\max}$ is set at 30° in this study; φ is the phase difference between heaving and rotating motions, set at π/2; f is the frequency; the motion period of the flapping hydrofoil T is 2 s, which according to the literature [20], to maximizes the discharge efficiency of suspended particles. These motion parameters were imposed on the reference node of the solid domain and transferred to Fluent through the System Coupling module at each coupling step, as detailed in Section 3.4.

In practical applications, to improve the operational stability of the flapping hydrofoil and avoid position errors caused by initial inclination angles, this study uses the extreme positions in the reciprocating motion cycle of the hydrofoil as the initial positions for motion. Figure 4 shows the complete cycle of hydrofoil motion.

### 2.3. Particle Phase Model and Discrete Phase Parameters

In subsequent numerical simulations, the Discrete Phase Model (DPM) in Ansys-Fluent is used to track the drift and sedimentation characteristics of suspended particles in the channel. The DPM based on the Lagrange method, tracks the movement trajectories of individual particles, with the particle force model considering drag, gravity, and buoyancy forces. Since the volume fraction of particles in this study is far less than that of water, and the continuous phase exerts significantly greater influence on particle dynamics than the discrete phase contributes to fluid modifications, the interaction between particles and their impact on the fluid are ignored in subsequent research. The established differential equation for particle motion is as follows:(2)$$\left\{\begin{array}{l} \frac{dv_{p}}{dt}=\frac{1}{\tau_{p}}(v_{f}-v_{p})+(1-\frac{\rho_{f}}{\rho_{p}})g\\ \frac{dx_{p}}{dt}=v_{p} \end{array}\right.$$
where $x_{p}$ is the particle’s velocity vector, with units m; $v_{p}$ is the particle’s velocity vector, with units m/s; $v_{f}$ is the fluid’s velocity vector, with units m/s; $\rho_{p}$ and $\rho_{f}$ are the particle density and fluid density, respectively, both with units kg/m^3^; g is the gravitational acceleration vector; The particle relaxation time $\tau_{p}$ is the characteristic time for a particle to adjust its velocity to local flow changes (units: s). For a spherical particle in the Stokes limit (Rep ≪ 1), the relaxation time is:(3)$$\tau_{p}=\frac{\rho_{p}d_{p}^{2}}{18\mu }$$
This expression follows the standard result for particle–flow dynamics originally proposed by Maxey and Riley [34]. Where $d_{p}$ is the particle diameter, with units m; μ is the fluid dynamic viscosity, with unit Pa⋅s. This expression follows the standard result for particle–flow dynamics originally proposed by Maxey and Riley [34]. For finite particle Reynolds numbers, we reference the Schiller–Naumann [35] drag correlation commonly used in CFD.

This study only considers the drift and sedimentation characteristics of particles, ignoring their evaporation, dissolution, and breakage. Before enabling the DPM, it is necessary to determine the main physical properties of particles in the discrete phase, including particle density, diameter, and total mass. Based on previous research on suspended particles in recirculating aquaculture systems [36,37,38], the selected particle density is $\rho_{p}=1100\;\mathrm{kg}/m^{3}$, and the average particle diameter is $d_{p}=0.5\;\mathrm{mm}$. Research by Zhang et al. [39] indicates high-density aquaculture water contains TPM levels of 31.58±3.38 mg/L (feed residues) and 29.75±5.26 mg/L (fish feces). These measurements were cross-referenced with the dimensional specifications of the experimental raceway. Combined with the raceway dimensions selected in this study, the calculated total particle mass is 1.2632 kg (feed residues) + 1.190 kg (fish feces) = 2.4532 kg.

## 3. Numerical Methods and Validation

### 3.1. Fluid Domain Control Equations

The flow of incompressible turbulence can be described by the following Reynolds-averaged Navier–Stokes equations and continuity equation [40]:(4)$$\left\{\begin{array}{l} \frac{\partial \bar{u_{i}}}{\partial t}+\bar{u_{j}}\frac{\partial \bar{u_{i}}}{\partial x_{j}}=-\frac{\partial \bar{P}}{\partial x_{i}}+\frac{\partial \cdot \left[\left(\upsilon +\upsilon_{T}\right)\left(\frac{\partial \bar{u_{i}}}{\partial x_{j}}+\frac{\partial \bar{u_{j}}}{\partial x_{i}}\right)\right]}{\partial x_{j}}\\ \frac{\partial \bar{u_{i}}}{\partial x_{i}}=0 \end{array}\right.$$
where $\bar{u_{i}},\bar{u_{j}}$ is the average fluid velocity in the i,j direction; $x_{i},x_{j}$ is the spatial coordinate in the i,j direction; Here $\bar{P}$ denotes the modified, density-normalized pressure:(5)$$\bar{P}=\frac{\bar{p}}{\rho_{f}}+\frac{2}{3}k$$
where $\bar{p}$ is the Reynolds-averaged physical pressure; k is the turbulent kinetic energy.

υ and $\upsilon_{T}$ are the molecular viscosity coefficient and eddy viscosity coefficient, respectively. For high Reynolds number turbulence, the eddy viscosity coefficient can be expressed as:(6)$$\upsilon_{T}=C_{\mu }k^{2}/\varepsilon$$
where ε is dissipation rate, respectively; $C_{\mu }$ is a dimensionless constant.

In the absence of a mean inflow, a motion-based Reynolds number is defined using a characteristic velocity that combines the heaving and pitching motions:(7)Re=Ucharcv
where c is the chord length of the hydrofoil and v is the kinematic viscosity of the fluid. The characteristic velocity $U_{char}$ is expressed as:(8)$$U_{char}=\sqrt{{\left(2\pi fA\right)}^{2}+{\left(r2\pi f\theta_{\max}\right)}^{2}}$$

For turbulence model selection, Fluent provides various turbulence models for simulating different fluid flow problems. Among them, the k−ε model is a widely used turbulence model. While turbulence modeling approaches like Large Eddy Simulation (LES) and Reynolds Stress Models (RSM) present different trade-offs, the k−ε model achieves optimal compromise between solution precision and computational demand, establishing its widespread utility in engineering applications. Among multiple versions of the k−ε model (such as the standard k−ε model), the Realizable k−ε model, through improved flow equations, maintains good numerical stability, especially under complex boundaries and changing flow fields; it shows stronger robustness in unstable flows (such as vortex shedding, boundary layer flow, etc.) and periodic flows (such as vibrations and periodic flows). Therefore, this paper adopts the Realizable k−ε model to solve the N-S equations.

### 3.2. Structural Domain Dynamic Equations

The deformation of flexible hydrofoils requires consideration of forces applied by the fluid on the solid structure. Transient dynamic analysis also needs to comprehensively consider the effects of inertia, damping, and time-varying loads. The movement of the structure can be described by the following equation:(9)$$M\ddot{q}+C\dot{q}+Kq=F_{ext}(t)+F_{fluid}(t)$$
where *M* is the mass matrix; *C* is the damping matrix; *K* is the stiffness matrix. The mass and stiffness matrices are assembled by ANSYS Transient Structural from the specified material and geometry (linear elastic). Structural damping is neglected in the present study, i.e., *C* = 0, because the foil motion is prescribed (heave/pitch) and the reported metrics are taken after periodic steady state is reached. $F_{ext}(t)$ is the external load vector, including gravity, elastic recovery force, and inertial force; $F_{fluid}(t)$ is the fluid force vector acting on the hydrofoil. The external load vector and fluid force vector act on the fluid–structure coupling surface of the hydrofoil in the form of distributed loads. The selected flexible material has a density of 1000 kg/m^3^, a Poisson’s ratio of 0.49, and an adjustable elastic modulus. To isolate the effect of stiffness on FSI-driven pumping, we model the foil as a nearly incompressible solid with constant density $\rho_{s}=1000\;\mathrm{kg}/m^{3}$ and vary the elastic modulus E as a control parameter to span flexible-to-rigid regimes. This keeps the mass ratio $m=\rho_{s}/\rho_{f}$ fixed and avoids coupling stiffness with inertia.

### 3.3. Particle Evaluation Metrics and Distribution Strategy

For collecting discharged particles, this study uses conventional filtration devices as an example, installing filters at the outlet of the aquaculture tank to filter out suspended particles from the water and collect them in a weighing container for periodic weighing. The particle escape rate is used as the evaluation standard, with the calculation formula as follows:(10)$$E_{R}=\frac{m_{e}}{m_{d}+m_{e}}\times 100\%$$
where $E_{R}$ is the particle escape rate, $m_{e}$ is the mass of escaped particles, and $m_{d}$ is the mass of particles in the fluid domain.

The current DPM implementation in Fluent restricts particle seeding to specific geometric patterns, preventing truly randomized spatial distribution across irregular domains; only random distribution within a fixed radius circle is possible. To simulate the uniform random distribution of particles such as feed residues and fish feces as closely as possible, this study adopts the distribution method of evenly covering circular areas proposed by Li et al. [41], where circular areas are tangent to each other without overlapping. The distribution method is shown in Figure 5, with a total of 36 circular areas, ranging from 2.6 m to 19.6 m from the outlet. The injection diameter of a single particle group is 1 m, and the discrete phase is introduced after achieving steady-state velocity conditions from the oscillating hydrofoil. The particle injection time is 5.0 s, using non-steady particle tracking.

### 3.4. Two-Way Fluid–Structure Coupling Method

For the joint solution of fluid control equations and structural control equations, a partitioned iterative solution method is adopted [42], which offers more flexibility in computational models, methods, and mesh scales. The core idea of this method is to solve the fluid domain and solid domain separately at each time step, exchanging data (such as forces and displacements) at the fluid–structure coupling interface, and achieving consistency between the fluid and solid solutions through iteration. The flow chart of the partitioned iterative solution method is shown in Figure 6.

On the ANSYS Workbench platform, Fluent is used for flow field analysis in fluid–structure coupling simulation, the Transient Structural module for structural analysis, and the System Coupling module for connecting the above two modules. Data transfer is established at the fluid–structure coupling interface in the System Coupling module, setting corresponding coupling time, coupling steps, time step size, and other information. The time step size plays a very important role in unsteady calculations, and its selection will be explained in detail in subsequent content. The fluid–structure coupling simulation system established in Workbench is shown in Figure 7.

### 3.5. Mesh Generation and Solution Setting

Considering that three-dimensional fluid–structure coupling simulation requires enormous computational resources, and according to the literature [43], two-dimensional numerical simulation results are basically consistent with experimental results in the study of hydrofoil propulsion performance. This study employs exclusively 2D numerical approaches for all simulations and computational analyses. Their reasonableness will be further explained in Section 3.8. When simulating the motion of flexible materials, the overset mesh methodology is typically employed to prevent negative cell volumes during grid deformation. However, Numerical experiments demonstrated that the proposed methodology cannot be effectively coupled with CFD’s DPM for particle-laden flow simulations. Therefore, this paper adopts dynamic mesh technology in Fluent, where when the solid structure deforms, the flow field receives displacement data from the solid deformation and reconstructs the mesh in the corresponding area using dynamic mesh. In the pond raceway application scenario, the flapping foil functions as a circulation pump, and the water flow is entirely driven by the motion of the foil, with no-slip boundaries. Therefore, no velocity inlet or incident wave was set in this study. The specific boundary conditions are summarized in Table 1. The computational domain utilizes a partitioned mesh system—an adaptive unstructured grid for capturing hydrofoil kinematics and a regular structured grid for pollutant generation analysis, significantly improving numerical efficiency. To minimize numerical calculation errors and improve the simulation accuracy of flow patterns near critical surfaces, mesh refinement with boundary layer divisions was implemented at both the hydrofoil edges and wall interfaces. This specialized grid configuration allowed for more precise resolution of the hydrodynamic interactions occurring in these high-gradient regions, with a first layer mesh thickness of 0.0001 m and a target y+ value of 1. The specific mesh division of the fluid domain is shown in Figure 8.

During mesh reconstruction, only the unstructured mesh in the hydrofoil movement area participates in reconstruction. Smoothing and Remeshing are set in Dynamic Mesh, with the spring constant factor set to 0. The hydrofoil wall is set as System Coupling, the hydrofoil movement area is set as Deforming, and other areas are set as Stationary. For local cell mesh reconstruction, size functions are enabled, with the maximum cell skewness set to 0.7, and the mesh is redrawn at each iteration step. The mesh reconstruction at different moments in a single period of hydrofoil motion is shown in Figure 9.

The continuous phase density used in the calculation is $\rho_{f}=1000\;\mathrm{kg}/m^{3}$, dynamic viscosity $\mu =1.003\times 10^{-3}\;\mathrm{Pa}\cdot s$. Velocity-pressure coupling employs the SIMPLE method, the first-order upwind scheme for the dissipation rate term, the second-order upwind scheme for the turbulent kinetic energy term, and iterative implicit calculation for time advancement. Iterative methods are used to solve fluid algebraic equations. When the residuals of all variables are less than 10^−5^, the iteration is considered to have reached the convergence criterion.

### 3.6. Mesh Independence Verification

In CFD numerical calculations, the number of mesh points directly affects the efficiency and results of numerical calculations. To ensure the accuracy of simulation results, this section verifies the independence of mesh quantity. This paper selected three sets of mesh numbers: 27,838, 45,846, and 98,096, as shown in Figure 10. For numerical simulation, comparing the variation curves of hydrofoil thrust under different mesh numbers. The curve shows significant oscillations at the beginning due to unconverted solutions, while after about 15 s, thrust changes exhibit periodic variations. Therefore, the curve portion after 15 s was selected, as shown in Figure 11. With 27,838 mesh points, the model calculation results show significant errors, while the results calculated with 45,846 and 98,096 mesh points are basically consistent. Considering the allocation of computational resources and calculation efficiency, 45,846 mesh points are selected for calculation in subsequent research.

### 3.7. Time Step Selection

The selection of time step size is extremely important in two-way fluid–structure coupling calculations. Excessive time step sizes often result in negative volume issues during dynamic mesh reconstruction, while too small-time step sizes lead to decreased simulation efficiency. To ensure that simulation results are not affected by time step size, under the premise of avoiding negative volumes that could cause solver errors, this paper compares the hydrofoil thrust coefficient variation curves with time step sizes of 0.004 s, 0.001 s, and 0.0005 s. The results are shown in Figure 12. The hydrofoil coefficient variation curves under the selected time step sizes are nearly identical. Therefore, in subsequent calculations, the time step size is set to 0.004 s.

### 3.8. Method Validation

To further verify the accuracy of numerical simulation experiments, this section validates the discrete phase and continuous phase separately. No new physical experiments were conducted in this study. The experimental data used for validation were obtained from published literature datasets.

Comparing the experimental results of the settling velocity of sediment particles in an organic glass settling tube from the literature [44], a settling model based on literature conditions was established. The settling tube dimensions are, sediment particle density 998.2 kg/m^3^, viscosity, calculating the settling velocities of sediment particles with diameters of 0.0002 m, 0.0004 m, 0.0006 m, 0.0008 m, 0.001 m. The results are shown in Figure 13. By comparison, although there are certain errors between experimental values and simulation values, the deviation values remain consistently below the 10% threshold. Therefore, it can be considered that the simulation method adopted in this paper is feasible and effective.

To assess the reliability of the fluid dynamics computations, results were cross-verified with available experimental datasets from prior studies [45]. Computations utilize the NACA0012 airfoil geometry with characteristic chord dimension c = 0.1 m, and calculations were performed for the instantaneous thrust coefficient $C_{T}$ under various working conditions with different St (Strouhal numbers, a dimensionless number) under the same maximum angle of attack, where U=0.4 m/s, ϕ=π/2, $A_{\max}=0.075\;m$, $\theta_{\max}=\pi /6$, Re=4×104, $St=2fA_{\max}/U$, and U represents the inlet velocity of the computational domain. The instantaneous thrust coefficient $C_{T}$ is calculated using the formula:(11)$$C_{T}=\frac{2F_{x}(t)}{\rho_{f}U^{2}c}$$
where $F_{x}(t)$ is the instantaneous thrust of the airfoil.

The results are shown in Figure 14. Experimental validation shows excellent correspondence with thrust coefficient predictions, verifying the reliability of the computational framework.

## 4. Results & Discussion

### 4.1. Effects of Hydrofoil Motion on Particulate Transport

This section first analyzes the impact of oscillating hydrofoils on the discharge of suspended particles in water under the combined heaving and pitching motion mode, establishing a foundation for subsequent comparative studies. Taking the flexible hydrofoil with an elastic modulus of 0.05 GPa as an example, post-processing techniques for simulation results are used to analyze the fluid characteristics in the flow field. Figure 15 shows the vorticity distribution cloud map after the flow field stabilizes, where no obvious backflow phenomenon is observed in the channel. The curved black arrows in the figure indicate the rotation direction of the vortices forming the reversed Kármán vortex street. Each motion cycle generates paired vortical structures: a negative (blue) vorticity field peeling from the lower foil contour, balanced by a positive (red) rotational structure shedding from the upper profile. The two separated vortices move toward the outlet direction with the fluid, and as the movement distance increases, their intensity gradually decreases until dissipation under the influence of gravity, evolving into a stable layered velocity distribution after dissipation. This pair of alternating, anisotropic vortices formed under the continuous motion of the hydrofoil is called a reversed Kármán vortex street. In the context of this study, suspended particle discharge rates scale with the vortex street’s structural coherence and persistence duration in the aquatic channel. Therefore, adjusting the elastic modulus of the hydrofoil to maximize the stability and persistence of the reversed Kármán vortex street is key to improving discharge efficiency.

Figure 16 shows the vorticity changes at different time points in a single hydrofoil cycle. It can be observed that when the hydrofoil begins to move upward, accompanied by counterclockwise rotation, a positive vortex, and a negative vortex are produced at the leading and trailing edges of the lower surface, respectively. The black circles in the figure mark the vortex cores to highlight the evolution and shedding process of the vortices during one motion cycle. Subsequently, the vortex at the leading edge moves toward the trailing edge along the hydrofoil surface during the motion, eventually forming a vortex with the trailing edge vortex when the hydrofoil moves downward. The trailing edge vortex separates from the hydrofoil when it reaches the extreme motion position, thus generating a pair of anisotropic vortices in a complete motion cycle, forming a reversed Kármán vortex street. Additionally, when vortices contact the channel walls, relatively larger vortices are produced, which can help suppress the sedimentation of particles at the bottom due to gravity.

Figure 17 shows the velocity distribution cloud map of discrete particles in 6000-time steps, with a time step size of 0.004 s and a total calculation time of approximately 168 h. Particle injection begins after 45 s when the flow field stabilizes, with an injection time of 5 s. In the initial injection period (45–49 s), the number of particles in the flow field continuously increases, reaching a maximum at around 50 s. In the early discharge period (45–53 s), the migration rate of particles in the middle of the channel is significantly higher than in near-wall regions. Additionally, the dissipation effect of the reversed Kármán vortex street generates a certain thrust along the vortex core, and the front end of the particle injection happens to be in an area with higher vortex street dissipation, causing particles in the front section of the channel to be quickly pushed to the middle section. As the advancement rate of particles in the middle section is less than in the front section, most particles accumulate in the middle section of the channel. In the middle discharge period (53–65 s), the particle distribution shows a typical vortex pattern, with the particle rates in the middle of the channel and near the walls gradually approaching, and the particle discharge efficiency tending to stabilize. In the late discharge period, the majority of particulate matter was successfully flushed through the system, with minimal gravitational deposition observed along the terminal section’s lower boundary.

### 4.2. Influence of Hydrofoil Elastic Modulus on Particle Discharge Efficiency

Figure 18 shows the deformation of hydrofoils with different elastic moduli at the same moment. A hydrofoil with an elastic modulus of 200 GPa undergoes almost no deformation and can be considered a rigid hydrofoil, serving as the control group. It can be observed that the deformation of various airfoils is mainly concentrated in the trailing edge area. Due to the constraints of dynamic mesh reconstruction, the range of elastic modulus variation selected in this study is relatively small, with limited overall hydrofoil surface deformation, especially in the 0.2 GPa group, where the deformation amplitude is close to that of the rigid hydrofoil. This also results in the overall similar trend in the change in particle escape rate over time under the action of hydrofoils with different elastic moduli shown in Figure 19, with the particle escape rate in all groups reaching a maximum value of about 97.8% after 69 s. Despite the similarity in final discharge efficiency, during the middle discharge period, the hydrofoil with the greatest deformation has a particle escape rate of 3–4% higher than the rigid hydrofoil. As the number of particles in the channel decreases, the time points at which various elastic modulus hydrofoils reach the maximum particle discharge rate tend to converge.

To further analyze this phenomenon, Figure 20 shows the slope changes in the escape rate curves for each group. In the early to middle discharge period, when most particles are still in the channel, the growth rate of particle discharge efficiency in the group with the lowest elastic modulus (most flexible hydrofoil) is significantly higher than in other groups; but as almost half of the particles are discharged, the efficiency of particle escape gradually decreases. Currently, the proportion of residual particles settling near the lower wall at the channel outlet is higher in this group than in other groups, reducing its discharge efficiency.

Figure 21 shows the vorticity cloud maps formed by hydrofoils with different elastic moduli at the same moment. By comparison, the first blue negative vortex formed at the trailing edge of the rigid hydrofoil detaches earlier, while in the flexible hydrofoil groups, except for the 0.2 GPa elastic modulus group, the vortex detachment time at the trailing edge is significantly delayed due to surface deformation in the remaining three groups. This is because the 0.2 GPa hydrofoil has a small deformation, making its overall cloud map like that of the rigid hydrofoil, while the relatively significant deformation at the trailing edge of the other three groups of hydrofoils causes a lag in flow field changes. The lag time increases with the increase in hydrofoil deformation, and this lag extends the time of tail vortex formation, leading to a delay in detachment time. Observing the shape of the last blue vortex in each group, it can be found that with the increase in material deformation, flexible hydrofoils release more energy when the tail vortex detaches, making the tail flow last longer, while the rigid hydrofoil’s tail vortex detaches earlier, with a more limited tail flow action range. This results in higher particle discharge efficiency for hydrofoils with greater deformation during the middle discharge period.

Figure 22 shows the channel velocity profile under the action of hydrofoils with different elastic moduli. It can be found that the water flow velocity through the reversed Kármán vortex street accelerates, presenting a jet-like state. In the first half of the flow field, the velocity stratification effect is obvious, which is conducive to the rapid migration of particles; in the second half of the flow field, the flow velocity tends to stabilize, reducing the discharge efficiency of particles. Comparing the positions of the 1.045 m/s isovelocity lines, with the increase in hydrofoil deformation, the vortex lasts longer, and the distance over which the stratified velocity structure is maintained gradually extends, especially for the 0.05 GPa flexible hydrofoil, where the maintained distance exceeds the entire channel, effectively improving the discharge efficiency of particles.

## 5. Conclusions

This paper, using the characteristic flow patterns of rectangular aquaculture raceways as a representative case, discusses the application of heaving and pitching flapping hydrofoil devices in aquaculture. Through two-way fluid–structure coupling numerical simulation, it comparatively analyzes the impact of flexible material hydrofoils versus rigid hydrofoils on the discharge of suspended particles in water. By adjusting the elastic modulus of hydrofoil materials in simulation comparison experiments, the following conclusions are drawn:The reversed Kármán vortex street generated during the motion of the flapping hydrofoil can effectively enhance the flow effects of raceway aquaculture water, promoting the discharge of suspended particles.Within the range of elastic moduli tested, materials with lower elastic moduli show higher discharge efficiency for particles in the early to middle discharge periods, but there is no significant difference in the final discharge effect, with the discharge rate of suspended particles in all groups at around 97.8%.Compared to rigid flapping hydrofoils, the surface deformation of flexible hydrofoils is mainly concentrated in the trailing edge portion. The deformation of the trailing edge delays the time of tail vortex detachment, increases the duration of the tail vortex, and extends the distance over which the stratified velocity structure is maintained.

The bioinspired design of flexible hydrofoils offers significant potential for improving pollutant discharge in aquaculture systems. By mimicking the dynamic adaptations of fish fins and bird wings, flexible hydrofoils can be optimized for greater efficiency in fluid management and environmental sustainability. However, in the present simulations the range of elastic moduli that could be explored was constrained by occasional negative cell volumes during dynamic mesh reconstruction under large flexible-surface deformations. As for the limitations, in real materials, the elastic modulus E and the solid density $\rho_{s}$ may co-vary; here, we deliberately decouple them (fixing $\rho_{s}$) to reveal stiffness-only trends, and we leave a systematic mapping over (E, $\rho_{s}$) for future work. To better probe and verify the discharge efficiency of highly compliant foils, future work will include experimental validation with flapping-hydrofoil devices, the use of materials permitting larger deformation in the hydrofoil profile, and exploration of integrating smart/active materials with bioinspired hydrofoils to further enhance adaptive capabilities in fluctuating fluid environments.

## Figures and Tables

**Figure 1 biomimetics-10-00779-f001:**
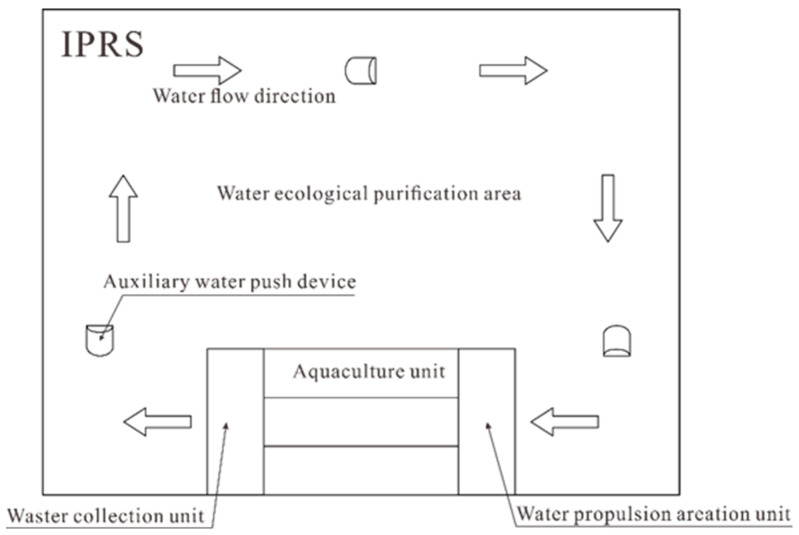
Visual overview of a recirculating aquaculture raceway design.

**Figure 2 biomimetics-10-00779-f002:**
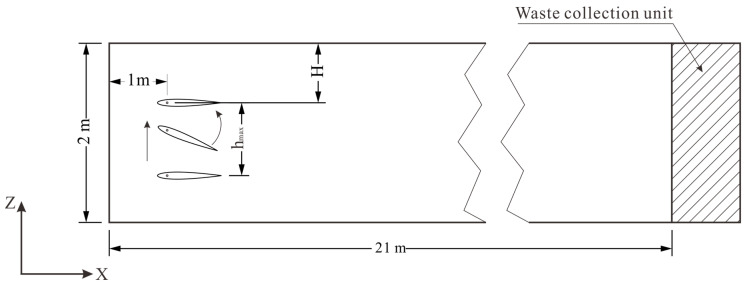
Side view of a single raceway.

**Figure 3 biomimetics-10-00779-f003:**
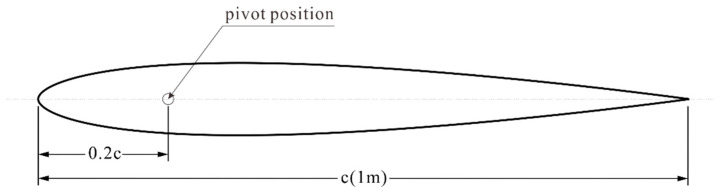
Dimensional diagram of NACA0012 hydrofoil.

**Figure 4 biomimetics-10-00779-f004:**
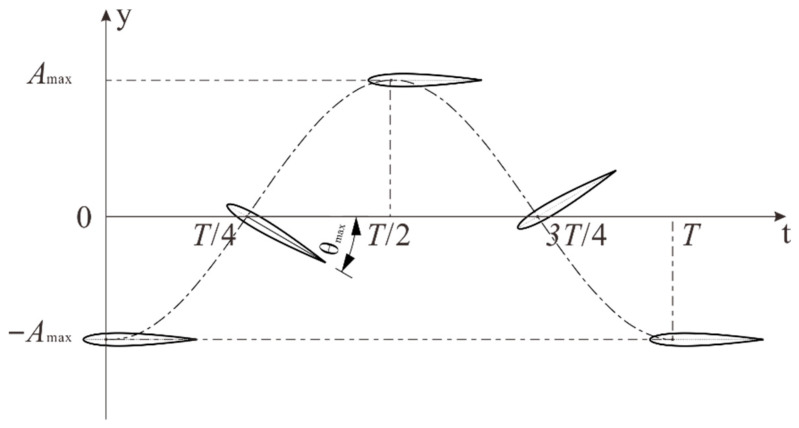
Diagrammatic representation of hydrofoil heaving and pitching motion.

**Figure 5 biomimetics-10-00779-f005:**
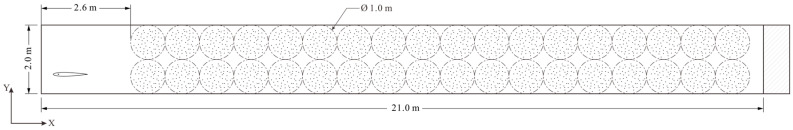
Schematic diagram of particle distribution strategy.

**Figure 6 biomimetics-10-00779-f006:**
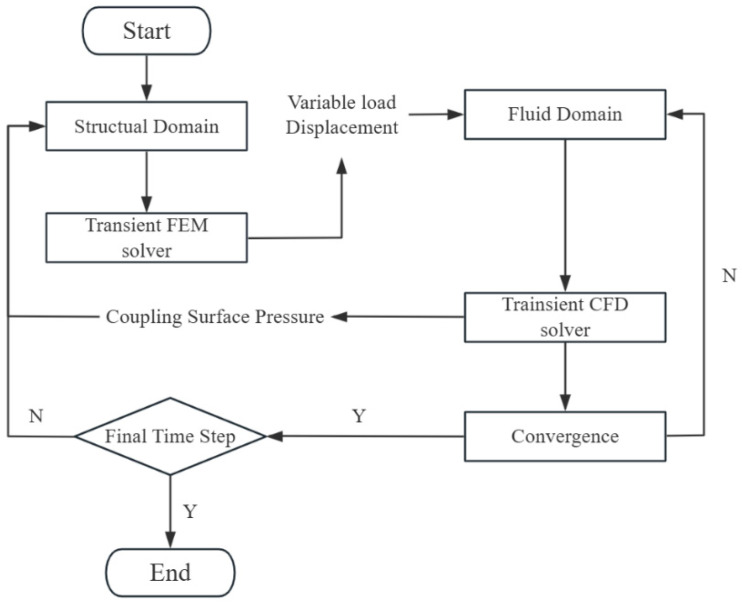
Computational flow chart.

**Figure 7 biomimetics-10-00779-f007:**
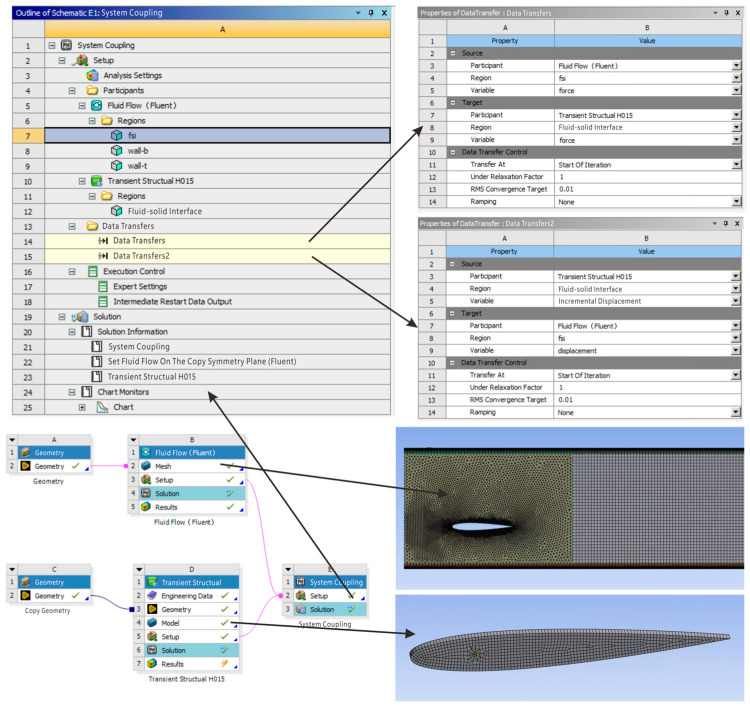
Two-way fluid–structure coupling simulation system.

**Figure 8 biomimetics-10-00779-f008:**
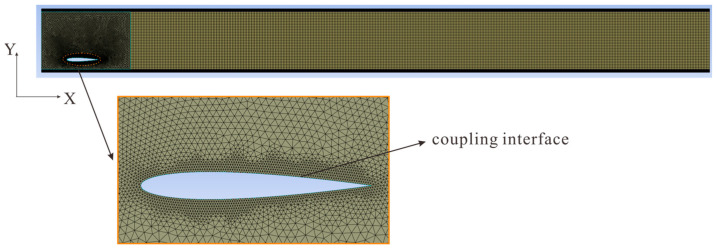
Mesh division diagram.

**Figure 9 biomimetics-10-00779-f009:**
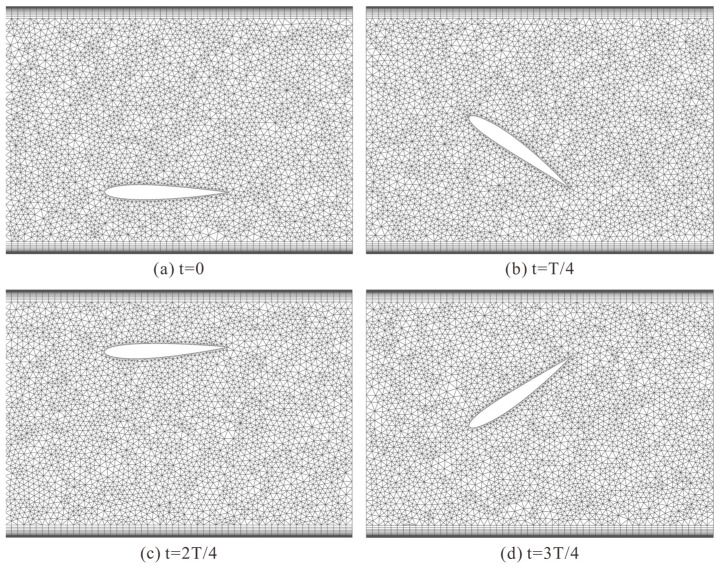
Mesh reconstruction schematic at different moments in a single period.

**Figure 10 biomimetics-10-00779-f010:**
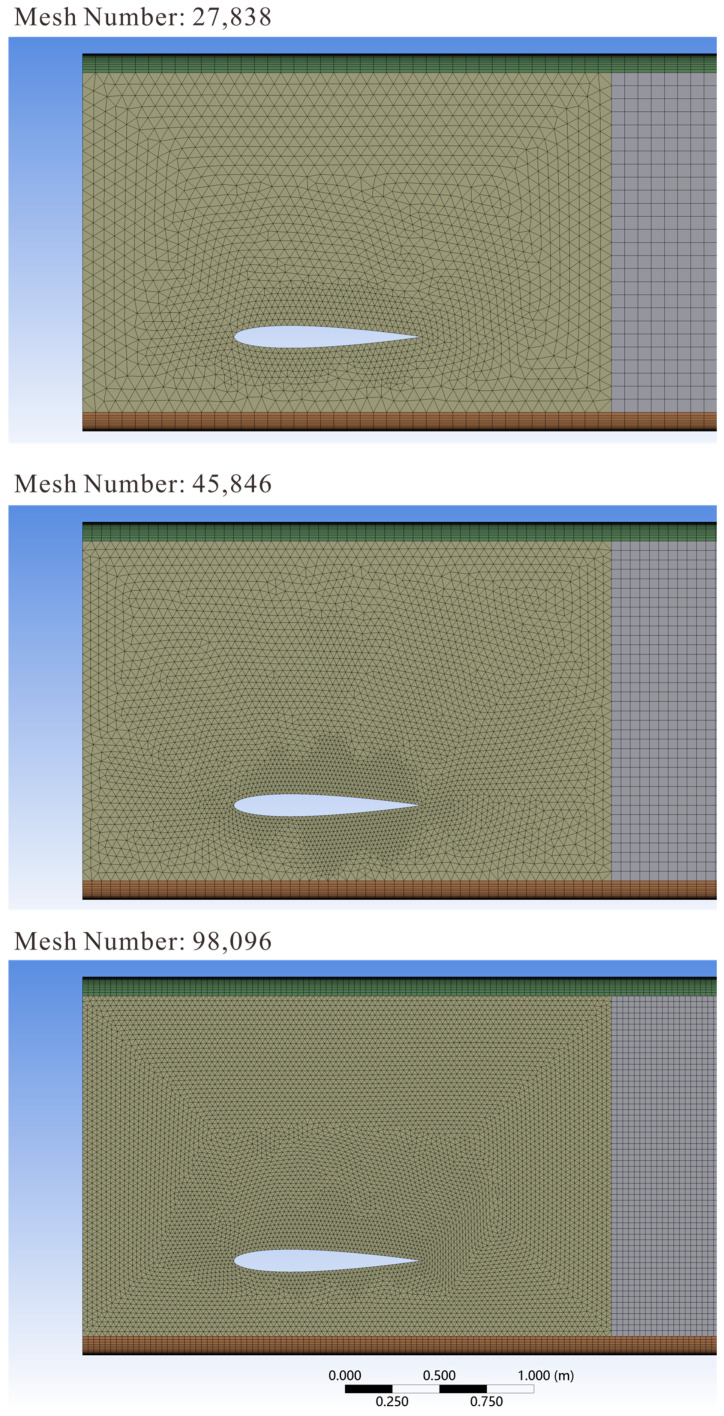
Comparison chart of different mesh sizes.

**Figure 11 biomimetics-10-00779-f011:**
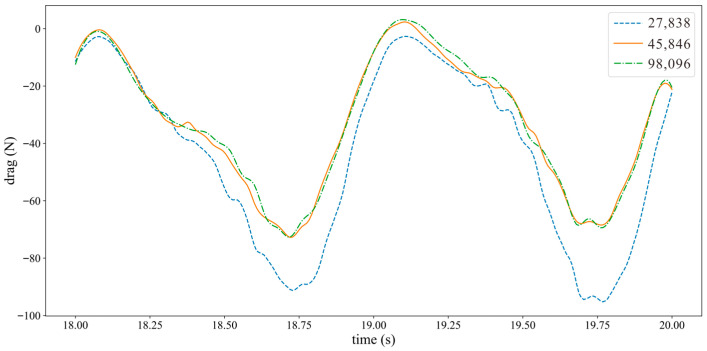
Mesh independence verification.

**Figure 12 biomimetics-10-00779-f012:**
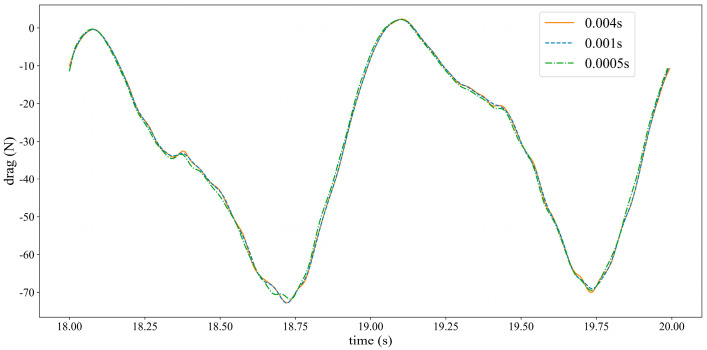
Time step independence verification.

**Figure 13 biomimetics-10-00779-f013:**
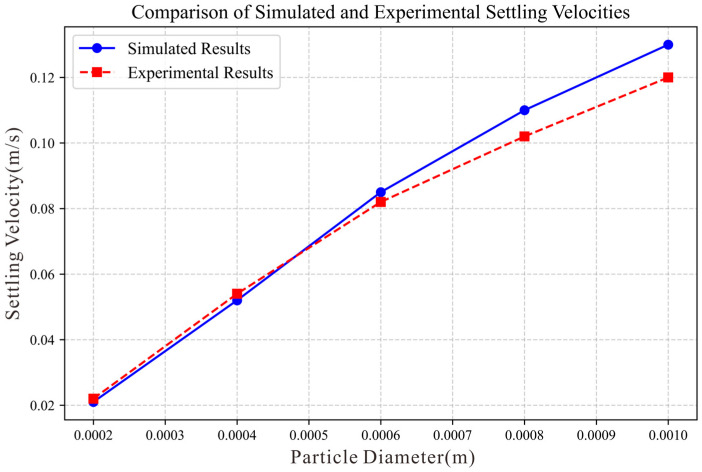
Comparison of simulated settling velocities of different particle sizes with experimental data.

**Figure 14 biomimetics-10-00779-f014:**
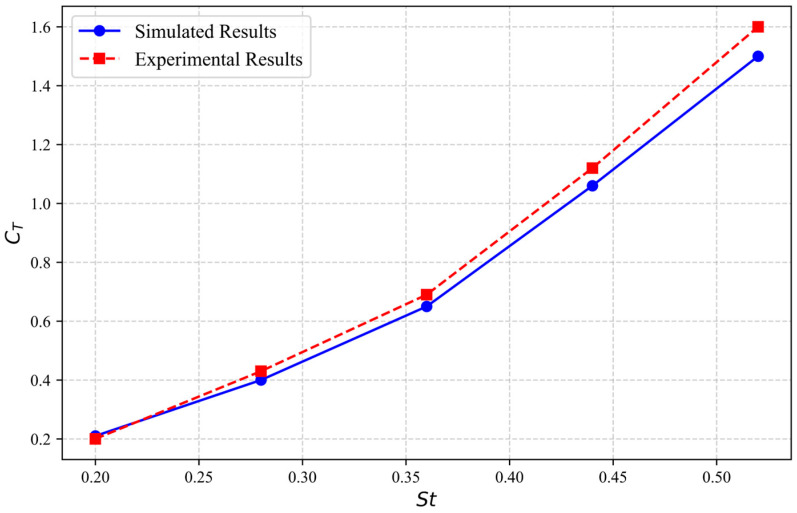
Comparison of numerical simulation calculation results with experimental data.

**Figure 15 biomimetics-10-00779-f015:**
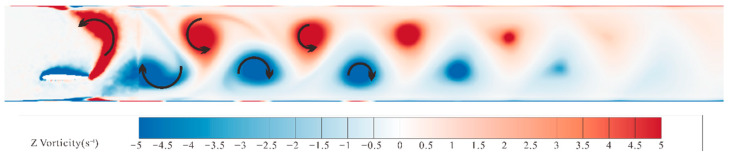
Vorticity cloud map of the stable flow field.

**Figure 16 biomimetics-10-00779-f016:**
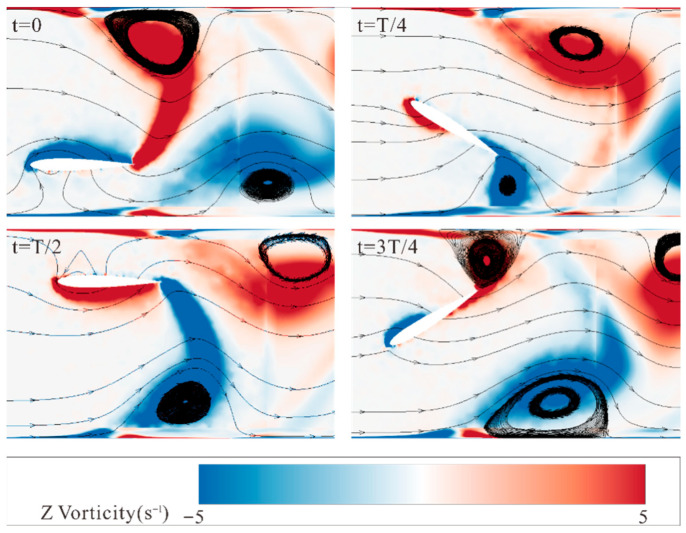
Vorticity changes the cloud map at different moments in a single cycle.

**Figure 17 biomimetics-10-00779-f017:**
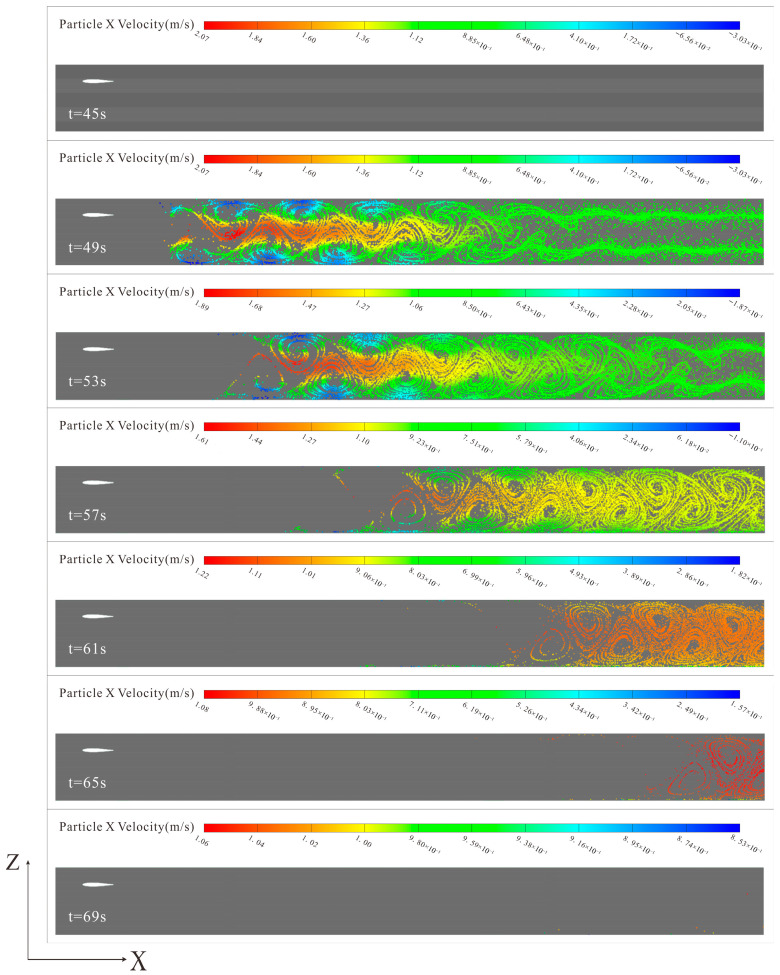
Particle distribution velocity cloud map at different moments in the channel.

**Figure 18 biomimetics-10-00779-f018:**
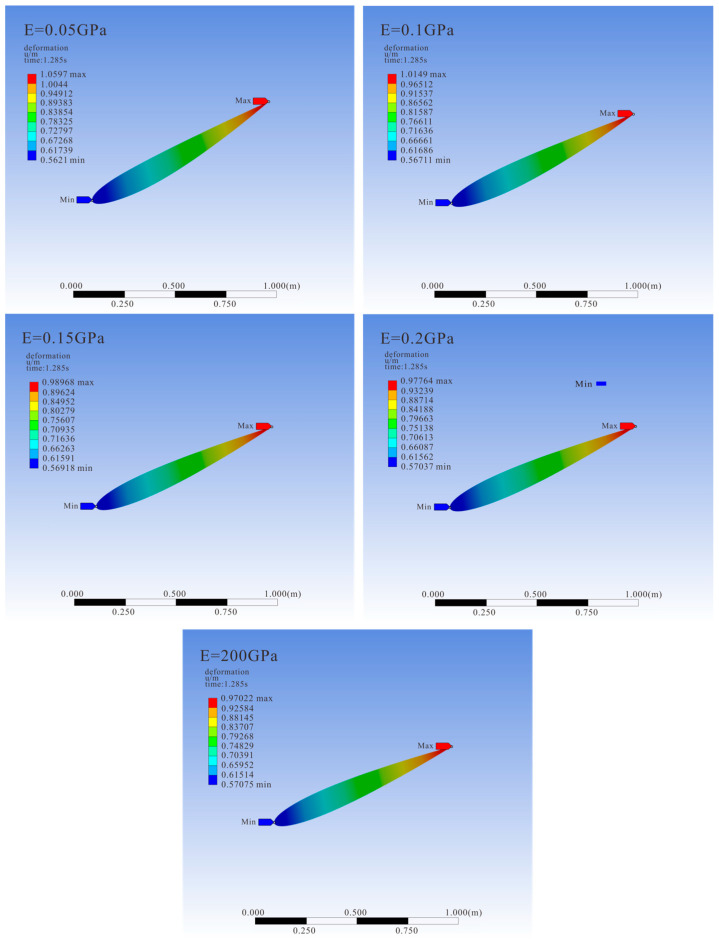
Equivalent elastic strain cloud map of hydrofoils with different elastic moduli.

**Figure 19 biomimetics-10-00779-f019:**
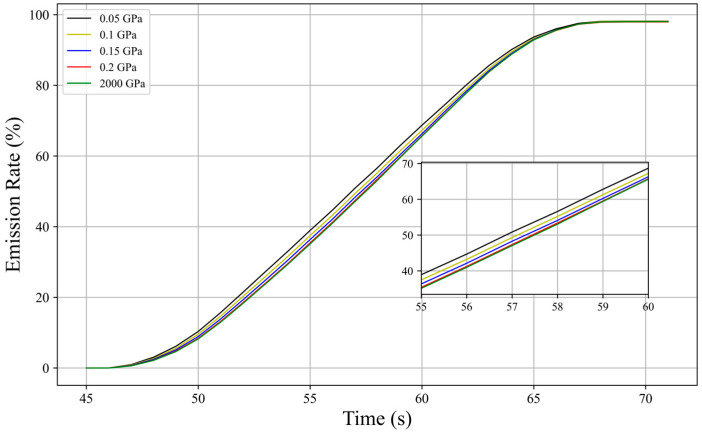
Particle escape rate curves for hydrofoils with different elastic moduli.

**Figure 20 biomimetics-10-00779-f020:**
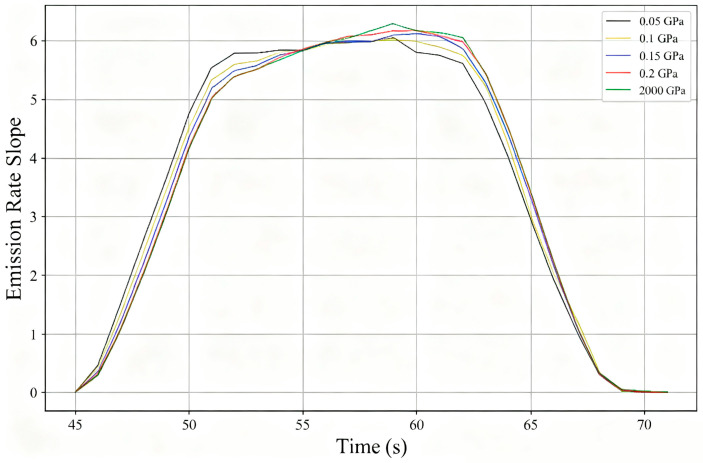
Particle escape rate slope curves for hydrofoils with different elastic moduli.

**Figure 21 biomimetics-10-00779-f021:**
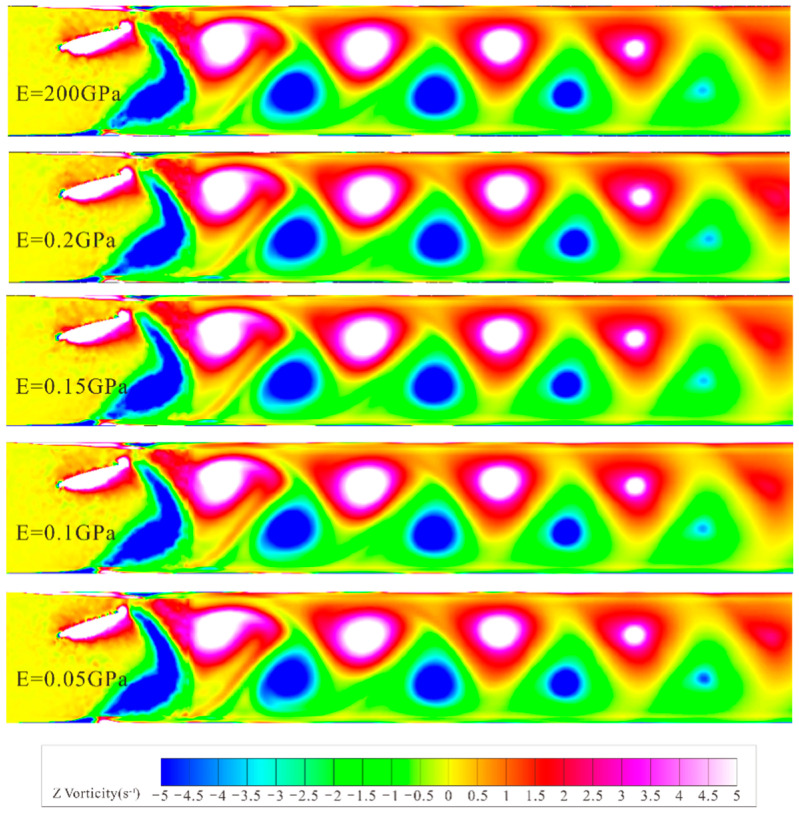
Vorticity cloud maps of hydrofoils with different elastic moduli.

**Figure 22 biomimetics-10-00779-f022:**
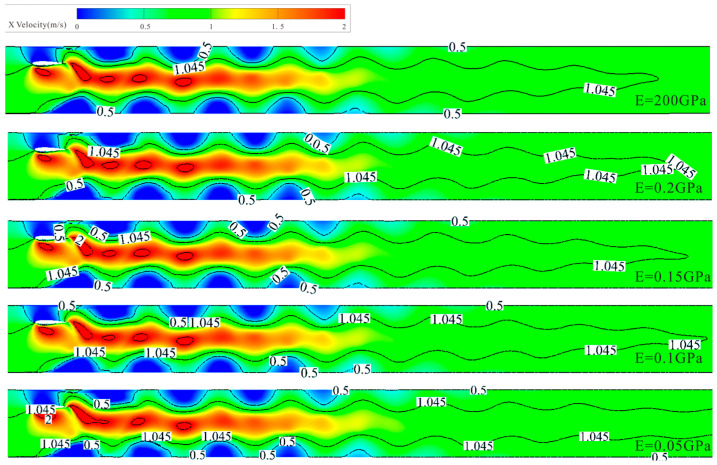
Flow velocity cloud maps of hydrofoils with different elastic moduli.

**Table 1 biomimetics-10-00779-t001:** Boundary conditions for the raceway case.

Boundary/Region	Type	Numerical Setting
Left boundary	Pressure inlet	Gauge *p* = 0 Pa; Backflow TI = 3.53%
Right boundary	Pressure outlet	Same as inlet
Top boundary	Wall	No-slip
Bottom boundary	Wall	No-slip
Hydrofoil surface	Moving wall via two-way FSI	Motion prescribed on structural side; transferred each time step
Particles (DPM)	Walls/outlet	Walls: reflect; Outlet: escape; Inlet: none

## Data Availability

The original contributions presented in this study are included in the article. Further inquiries can be directed to the corresponding author.

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
