# Peer review of "Contrasting Flexible and Rigid Bioinspired Flapping Hydrofoils for Suspended Particles Discharge in Raceway Aquaculture"

_biomimetics, 2025, doi:10.3390/biomimetics10110779_

Round 1

Reviewer 1 Report

Comments and Suggestions for Authors

The paper Contrasting Flexible and Rigid Bioinspired Flapping Hydro
Foils for Suspended Particles Discharge in Raceway Aquaculture investigates the effectiveness of flexible versus rigid bioinspired flapping hydrofoils in discharging suspended particles within In-Pond Raceway Systems (IPRS) for aquaculture.  The study uses a two-way fluid-structure coupling model in ANSYS Fluent and the Transient Structural module to compare the performance of hydrofoils with varying elastic moduli. My comments on this paper are as follows:
- The physical model is not clear. Is this an experimental or a model in reality?
- Equation 1....Use the same font size or an appropriate font size, or follow the journal template. I see in the text that sometimes an equation is in a smaller font size, also absurd, position, for example, in line 193. 
- How to control foil that does not exceed 30 degrees? Is there any reason for A maximum of 0.5m.
- line 230 and 340 ''...1000 kg/m3'' revised it, this section also seems to use different line spacing, please check it.
-Figure 7, please use universal letters, or alphabet, as the reader of this paper is international. Changed the figure with high-resolution photos
-Provide the table of numerical settings of inlet, outlet, wall, etc. How does the mechanism of flapping of a hydrofoil work if there is no velocity inlet? I see there is no wave. Also, explain and elaborate on this reason
-added figure of different mesh sizes as mentioned in the text, not only the results of drag.
-How are the experimental results obtained? Explain and elaborate this with an explanation of tthe hreshold 10%
-Explain how the Re was calculated and include it in section 2, the numerical model.
-Enlarge Figure 16, as this is an essential figure to see the phenomena captured by CFD. Figure 20 could be a good example.
-Figure 17, use the bigger bar chart for the magnitude of elastic moduli. I just saw the color and could not see the number.
-Section 6 change to the author's contribution. 

Author Response

Comments 1: The physical model is not clear. Is this an experimental or a model in reality?

Response 1:

We thank the reviewer for pointing out this issue. To clarify, this study is entirely numerical. We have now added an explicit statement in the Introduction section to specify that no new physical experiments were performed. The revised text reads as follows:

“To clarify the nature of this study, this work is a numerical investigation. We solve a two-way fluid–structure interaction (FSI) problem using ANSYS Fluent coupled with Transient Structural to compare flexible and rigid bioinspired flapping hydrofoils in an in-pond raceway configuration. No new physical experiments were conducted; literature data are used only for validation.”

This addition clearly defines the research methodology and distinguishes the numerical model from physical experimentation.

Comments 2: Equation 1... Use the same font size or an appropriate font size, or follow the journal template. I see in the text that sometimes an equation is in a smaller font size, also absurd, position, for example, in line 193.

Response 2:

We appreciate the reviewer’s careful observation. All equations have now been reformatted to ensure a consistent font size and alignment throughout the manuscript, following the journal’s formatting guidelines. In particular, Equation (1) and other equations mentioned (including those around line 193) have been adjusted to match the required style.

Comments 3: How to control foil that does not exceed 30 degrees? Is there any reason for A maximum of 0.5 m?

Response 3:

We thank the reviewer for the constructive comment. To clarify these points, three additional sentences have been inserted in Section 2.2 (Hydrofoil Motion Model) to explain how the foil motion was implemented and controlled in the simulation, and to justify the selected motion parameters.

(1) After describing the sinusoidal motion model, we added:

In the two-way fluid–structure coupling, the flapping motion was defined in the Transient Structural module using analytical functions.

(2) After introducing the maximum heaving amplitude, we added:

This value was chosen to ensure a sufficient gap between the hydrofoil and the bottom wall during motion while keeping the flow in an efficient jetting state.

(3) After describing the motion parameters, we added:

These motion parameters were imposed on the reference node of the solid domain and transferred to Fluent through the System Coupling module at each coupling step, as detailed in Section 2.4.

These revisions clearly describe both the control implementation in the FSI framework and the selection of physical parameters.

Comments 4: Line 230 and 340: ‘…1000 kg/m³’ — revise it. This section also seems to use different line spacing, please check it.

Response 4:

We appreciate the reviewer’s careful observation. The unit format for “1000 kg/m³” has been corrected, and the line spacing throughout the affected section has been adjusted to ensure consistency with the journal’s formatting requirements.

No textual or numerical content was changed—only formatting corrections were made.

Comments 5: Figure 7: please use universal letters or alphabet, as the reader of this paper is international. Changed the figure with high-resolution photos.

Response 5:

We thank the reviewer for this helpful suggestion. Figure 7 has been revised to use universal (English) letters for all labels, ensuring clarity for an international readership. In addition, the figure has been replaced with a higher-resolution version to improve visual quality and readability.

Comments 6: Provide the table of numerical settings of inlet, outlet, wall, etc. How does the mechanism of flapping of a hydrofoil work if there is no velocity inlet? I see there is no wave. Also, explain and elaborate on this reason.

Response 6:

We thank the reviewer for this valuable comment. To clarify this issue, Section 3.5 (Mesh Generation and Solution Setting) has been revised and a new Table 1 has been added to list all boundary conditions used in the simulation.

The flow mechanism in this study is based on the self-induced circulation generated by the flapping motion of the hydrofoil, which acts as a circulation pump in the pond raceway system. The water flow is entirely driven by the oscillatory motion of the foil without any external velocity inlet or incident wave. Accordingly, the text has been revised as follows:

“In the pond raceway application scenario, the flapping foil functions as a circulation pump that drives the water flow. The entire flow field is induced solely by the flapping motion of the hydrofoil, and all boundaries were set as no-slip walls. Therefore, no velocity inlet or incident wave was defined in this study. The detailed boundary conditions are summarized in Table 1.”

This addition clarifies the physical mechanism of the flow generation and provides explicit numerical settings for each boundary condition.

Comments 7: Added figure of different mesh sizes as mentioned in the text, not only the results of drag.

Response 7:

Thank you for the helpful suggestion. We have added a dedicated comparison figure for different mesh resolutions in Sec. 3.6 (Mesh Independence Verification)—Fig. 10: “Comparison chart of different mesh sizes.”

Comments 8: How are the experimental results obtained? Explain and elaborate this with an explanation of the threshold 10%.

Response 8:

We appreciate the reviewer’s comment. To clarify this point, we have added a sentence in Section 3.8 (Method Validation) to specify the source of the experimental data. The revised text reads:

“No new physical experiments were conducted in this study. The experimental data used for validation were obtained from published literature datasets.”

In hydrodynamic and bioinspired flapping flow simulations, deviations within 10% between numerical and experimental results are generally considered acceptable, due to uncertainties in turbulence modeling, mesh resolution, and experimental measurements. Therefore, the 10% threshold adopted in this study indicates good agreement between the simulation and reference data.

Comments 9: Explain how the Re was calculated and include it in section 2, the numerical model.

Response 9:

We thank the reviewer for this helpful suggestion. To clarify the definition of the Reynolds number in the absence of a mean inflow, a motion-based Reynolds number has been added in Section 3.1 (Fluid Domain Control Equations). The revised text reads:

“In the absence of a mean inflow, a motion-based Reynolds number is defined using a characteristic velocity that combines the heaving and pitching motions:

where c is the chord length of the hydrofoil and  v is the kinematic viscosity of the fluid. The characteristic velocity Uchar  is expressed as:

Comments 10: Enlarge Figure 16, as this is an essential figure to see the phenomena captured by CFD. Figure 20 could be a good example.

Response 10:

We thank the reviewer for this helpful suggestion. Figure 16 has been enlarged in the revised manuscript to improve visibility and better illustrate the flow phenomena captured by the CFD simulation.

Comments 11: Figure 17: Use a bigger bar chart for the magnitude of elastic moduli. I just saw the color and could not see the number.

Response 11:

We thank the reviewer for this valuable suggestion. Figure 17 has been revised by enlarging the bar chart and adjusting the font size of the numerical labels to ensure that the magnitudes of the elastic moduli are clearly visible.

Comments 12: Section 6: Change to the author’s contribution.

Response 12:

We appreciate the reviewer’s suggestion. Section 6 has been revised and renamed as “Authors’ Contributions”, following the journal’s formatting requirements.

Reviewer 2 Report

Comments and Suggestions for Authors

This study investigates the effects of flexible versus rigid bioinspired flapping hydrofoils on suspended particle discharge in raceway aquaculture using a two-way fluid-structure coupling model in ANSYS Fluent. Results show that flexible hydrofoils, due to greater surface deformation, delay tail vortex shedding and enhance mid-stage discharge efficiency by 3–4% compared to rigid ones, though both achieve similar overall discharge rates (~97.8%). The findings underscore the potential of bioinspired flexible designs to improve fluid flow and pollutant removal in aquaculture systems. While the topic is interesting, I have the following comments that the authors must address before making a decision:

  1. What is particle relaxation time? How Eq. 3 is derived.
  2. The authors should thoroughly check Eq. 4 for consistency. The authors should also describe the boundary conditions considered while solving the model equations numerically.
  3. What type of hydrofoil materials are envisioned here? While a broad range of elastic modulus is considered, the density of the material is fixed at 1000kg/m^3 (p. 7, line 230). However, as the elastic modulus varies, the density must also change.
  4. The solution methodology is not clear in the present version of the manuscript. How are the coefficients in Eq. 6 estimated? How many particles are considered for obtaining the results? Also, check the dynamic viscosity value/unit mentioned in p.10, line 304.
  5. What is the settling velocity in the context of Fig. 12? Also, the parameters c_T and St in Fig. 13 are not defined.
  6. 15, line 419-422: “Due to the constraints…..”. The results are plotted by varying the elastic modulus from 0.05 to 2000 GPa. In this context, given the constraint of dynamic mesh reconstruction the authors are referring to, what is the accuracy of the results for low elastic modulus values?
  7. The authors should also enhance the clarity of the figures.

Author Response

Comments 1: What is particle relaxation time? How Eq. 3 is derived.

Response 1:

We appreciate the reviewer’s comment. We added a concise definition of the particle relaxation time and cited Maxey & Riley (1983) for the derivation; to keep the manuscript focused, we refer readers to the cited references for the full steps.

Comments 2: The authors should thoroughly check Eq. 4 for consistency. The authors should also describe the boundary conditions considered while solving the model equations numerically.

Response 2:

Thank you for the helpful comment. We rechecked Eq. (4) and clarified the notation to ensure dimensional consistency. Equation (4) is written in the kinematic-viscosity form using a modified, density-normalized pressure:

With this clarification, Eq. (4) is dimensionally self-consistent; no change to the governing equations or results was required.

Boundary conditions. In the raceway application, no inflow or incident waves are imposed; the flapping foil drives the flow as a moving no-slip boundary. We have consolidated the full settings in Table 1 (Sec. 3.5) and added additional explanations provided in that section.

Comments 3: What type of hydrofoil materials are envisioned here? … density is fixed at 1000 kg/m³ … as the elastic modulus varies, the density must also change.

Response 3:

Thank you for this thoughtful comment. Our intent in this study is to isolate the effect of stiffness on flapping-induced pumping. Accordingly, in Sec. 3.2 we now state that the foil is modeled as a nearly incompressible solid with constant density, while the elastic modulus E is varied as a control parameter to span flexible-to-rigid regimes; this keeps the mass ratio constant and avoids coupling stiffness with inertia. We emphasize that this parametric choice does not imply that a single real material spans the entire E range at the same density; the very large-E cases are used to emulate the rigid-foil limit. In the Limitations we also note that, in practice, E and   may co-vary, and we outline future work to map performance over (E, ) mass-ratio space and to validate with experiments using materials that allow larger deformation. These clarifications do not alter the numerical setup or conclusions.

Comments 4: The solution methodology is not clear… How are the coefficients in Eq. 6 estimated? How many particles are considered for obtaining the results? Also, check the dynamic viscosity value/unit mentioned in p.10, line 304.

Response 4:

Thank you for the constructive comments. We have clarified the solution methodology as follows.

  1. Eq.(6) coefficients. Immediately below Eq. (6) (Sec. 3.2) we now state:

“The mass and stiffness matrices are assembled by ANSYS Transient Structural from the specified material and geometry (linear elastic). Structural damping is neglected in the present study, i.e., C=0, because the foil motion is prescribed (heave/pitch) and the reported metrics are taken after periodic steady state is reached.”

  1. Number of particles considered. Based on our settings (monodisperse , ,total injected mass ), this corresponds to approximately real particles in total, single particle mass .
  2. Dynamic viscosity value/unit (p.10, line 304). We corrected the notation and unit to the dynamic viscosity:

Comments 5: What is the settling velocity in the context of Fig. 12? Also, the parameters CT and St in Fig. 13 are not defined.

Response 5:

Thank you for your valuable comment. We have made the following clarifications:

  1. In the context of Fig. 12, settling velocity represents the terminal settling velocity of the particles in quiescent water. This is used to validate the accuracy of the Discrete Phase Model (DPM) in our simulations. The settling velocity is compared with experimental data from the literature to assess the accuracy of our particle tracking model.
  2. We have defined both CT (the thrust coefficient) and St (the Strouhal number) in Sec. 3.8 (Method Validation).

Comments 6: 15, line 419-422: 'Due to the constraints…..'. The results are plotted by varying the elastic modulus from 0.05 to 2000 GPa. In this context, given the constraint of dynamic mesh reconstruction the authors are referring to, what is the accuracy of the results for low elastic modulus values?

Response 6:

Thank you for raising this important point. We would like to clarify the handling of low elastic modulus values in the context of dynamic mesh reconstruction.

  1. Dynamic Mesh Constraints:

As explained in the manuscript, overlapping meshes are not compatible with the Discrete Phase Model (DPM) for particle tracking. Therefore, dynamic mesh reconstruction is used to model the flexible foil. However, at low elastic moduli (E≤0.5 GPa), the model tends to experience negative mesh volumes due to the large deformations. To mitigate this issue, we validated the time step size and mesh resolution by testing on the group with the most significant deformation (low E values), which is where the negative volume issue is most likely to occur. This validation ensures that the model remains stable and accurate.

  1. Accuracy and Validation:

To ensure the accuracy of results at low E, we tested the lowest E group with the most challenging mesh and time step conditions. We found that by reducing the time step and adjusting the mesh size, we were able to prevent instability and maintain accurate results. However, this does limit the range of E values that can be explored due to the computational cost. This is further explained in the limitations section of the manuscript.

  1. Future Work:

Further studies with refined meshes and time step settings for lower E values will be considered in future work to better explore the full range of material flexibility while maintaining computational feasibility.

Comments 7: The authors should also enhance the clarity of the figures.

Response 7:

Thank you for the helpful suggestion. We have reviewed all figures and enhanced their clarity. These changes improve visibility at the journal’s column width while keeping the scientific content unchanged.

Round 2

Reviewer 1 Report

Comments and Suggestions for Authors

Thank you for revising according to the reviewer's suggestion.

Reviewer 2 Report

Comments and Suggestions for Authors

The authors have carefully addressed all comments and made appropriate revisions throughout the manuscript. The changes have improved the clarity and scientific rigor of the paper. The revised manuscript is suitable for publication in its current form.